# Trapping Charge Mechanism in Hv1 Channels (*Ci*Hv1)

**DOI:** 10.3390/ijms25010426

**Published:** 2023-12-28

**Authors:** Miguel Fernández, Juan J. Alvear-Arias, Emerson M. Carmona, Christian Carrillo, Antonio Pena-Pichicoi, Erick O. Hernandez-Ochoa, Alan Neely, Osvaldo Alvarez, Ramon Latorre, Jose A. Garate, Carlos Gonzalez

**Affiliations:** 1Centro Interdisciplinario de Neurociencia de Valparaíso, Universidad de Valparaíso, Valparaíso 2351319, Chile; miguel.fernandez@postgrado.uv.cl (M.F.); juan.alveara@postgrado.uv.cl (J.J.A.-A.); christian.carrillo@cinv.cl (C.C.); antonio.pena@cinv.cl (A.P.-P.); alan.neely@uv.cl (A.N.); oalvarez@uchile.cl (O.A.); ramon.latorre@uv.cl (R.L.); 2Millennium Nucleus in NanoBioPhysics, Universidad de Valparaíso, Valparaíso 2351319, Chile; 3Cell Physiology and Molecular Biophysics, Texas Tech University Health Sciences Center, Lubbock, TX 79430, USA; emerson.rojas@ttuhsc.edu; 4Department of Biochemistry and Molecular Biology, University of Maryland School of Medicine, Baltimore, MD 21201, USA; ehernandez-ochoa@som.umaryland.edu; 5Facultad de Ingeniería, Arquitectura y Diseño, Universidad San Sebastian, Santiago 7780272, Chile

**Keywords:** *Ciona intestinalis*, proton channel, gating currents, charge trapping

## Abstract

The majority of voltage-gated ion channels contain a defined voltage-sensing domain and a pore domain composed of highly conserved amino acid residues that confer electrical excitability via electromechanical coupling. In this sense, the voltage-gated proton channel (Hv1) is a unique protein in that voltage-sensing, proton permeation and pH-dependent modulation involve the same structural region. In fact, these processes synergistically work in concert, and it is difficult to separate them. To investigate the process of Hv1 voltage sensor trapping, we follow voltage-sensor movements directly by leveraging mutations that enable the measurement of Hv1 channel gating currents. We uncover that the process of voltage sensor displacement is due to two driving forces. The first reveals that mutations in the selectivity filter (D160) located in the S1 transmembrane interact with the voltage sensor. More hydrophobic amino acids increase the energy barrier for voltage sensor activation. On the other hand, the effect of positive charges near position 264 promotes the formation of salt bridges between the arginines of the voltage sensor domain, achieving a stable conformation over time. Our results suggest that the activation of the Hv1 voltage sensor is governed by electrostatic–hydrophobic interactions, and S4 arginines, N264 and selectivity filter (D160) are essential in the *Ciona*-Hv1 to understand the trapping of the voltage sensor.

## 1. Introduction

The voltage-gated proton channel Hv1 is a membrane protein found in different kingdoms and its function varies depending on the species. For example, in dinoflagellates, it is related to bioluminescence [1]. In mammals, Hv1 is mainly associated with the immune system since it is functionally coupled with NADPH oxidase 2 (NOX2). NOX2 is related to processes such as histamine production by basophils and eosinophils, respiratory burst, and B lymphocyte signaling [2,3,4,5,6]. It has also been reported that, in some types of cancers, Hv1 presence seems to promote tumor invasion and migration, both processes regulated by intra and extracellular pH [7,8]. In general, Hv1 is present in processes where pH regulation is crucial for sustaining physiological functions. In terms of its structure–function relationship, little progress has been made compared to the diverse body of knowledge regarding its physiological function. Hv1 functions as a homodimer with monomers composed of four transmembrane segments. The S4 segment contains three arginines responsible for its voltage sensing [9,10]. Deleting part of its C- and N-terminus produces a fully functional monomeric channel (ΔNΔC) which preserves all the features of the native dimeric channel: high proton selectivity, voltage dependence and ΔpH sensitivity [11,12,13,14]. In addition, most of the Hv1 variants have an aspartate that is crucial for proton selectivity (D160 in *Ciona intestinalis* (*Ci*Hv1) and D112 in human (hHv1)) [11,12,13,14]; mutations in this amino acid allow the channel to permeate cations including guanidinium [11]. The substitution of an asparagine in the selectivity filter in ΔNΔC *Ci*Hv1 drastically reduces proton conduction through the channel in such a way that the charge-movement associated with voltage-sensor activation and deactivation (i.e., gating currents) can be detected [15]. In addition, gating currents can be modulated by the proton concentration gradient across the plasma membrane (ΔpH). When this gradient is positive (pH_o_ − pH_i_ > 0), a leftward shift of the Q(V) curve is obtained. This change in the resting–active equilibrium is a direct consequence of the energy stored by the proton’s gradient along the permeation pathway [15]. The voltage sensor displacement from a resting state to an active state dissipates this pH gradient, and the energy stored is strongly coupled to the voltage sensor allowing the channel to be activated with less electrical energy [15]. As previously reported, the N264R mutant exhibited a significant decrease in proton conductance compared to the wild-type channel [16,17]. This decrease in proton currents in the ΔNΔC N264R mutant allowed the detection of gating currents at the onset of the depolarization pulse; however, this mutant shows a reduction in the OFF-gating current when the membrane is repolarized, which reflects the movement of the voltage sensor to its deactivated state [16]. It is unclear whether WT channels also have a slow rate of voltage-sensor deactivation because the mutation at position 264 can be altering the native properties of the channel.

The molecular origin of the charge trapping of the Hv1 voltage sensor is not clear. This process could be a feature of its structure or, in principle, a feature due to the N264R mutation. The main aim of the present study is to unveil the molecular mechanism that underlies the gating charge trapping in Hv1 channels and how it relates to proton conduction and pH sensing. For this, we took advantage of the ΔNΔC D160N mutant where the ON-gating charge was recovered in the OFF-gating current component. Utilizing the ΔNΔC D160N N264R double mutant, we show that the presence of the arginine in position 264 partially restores the voltage sensor trapping in the D160N background. This suggests that the selectivity filter (D160) located in the transmembrane segment S1 in front of the voltage sensor is also involved in the movement of the voltage sensor. Therefore, we made multiple amino acid substitutions at position D160 and found that the more hydrophobic the mutation, the higher the activation energy required by the channel to activate. This is because the selectivity filter is a crucial element in the energy barrier that allows the movement of the gating particles.

On the other hand, to understand how the arginine in position 264 and the selectivity filter D160 affect the movement of the voltage sensor, we use the positively charged blocker 2-guanidinebenzoimidazole (2GBI). This blocker is a compound that strongly resembles an arginine and a benzimidazole ring [18,19]. Specifically, 2GBI possesses a guanidinium group in its molecular structure, identical to that of the arginine side chain. In addition, aromatic rings also provide a hydrophobic component to the molecule. It is known that the binding site of this classical Hv1 proton channel inhibitor involves the interaction between the selectivity filter (D160), the benzimidazole group and the guanidinium group with one of the S4 transmembrane arginines [18,19]. Thus, when 2GBI is added onto the intracellular side in the ΔNΔC D160N, it is possible to appreciate that the voltage sensor stabilizes in its active conformation. Moreover, the OFF-gating charge effect rivals the voltage sensor trapping observed in the ΔNΔC N264R variant.

It was also found that pH differences modulated the movement of the sensor as well, and the larger the pH gradient, the faster the sensor activation [15]. This pH gradient (ΔpH > 0) can also modify the deactivation of the voltage sensor by an evident decrease in the OFF-gating charge, suggesting that when there is a very high proton concentration on the intracellular side, they generate a local positive charge density around the 264 position, causing an effect similar to the N264R mutation and 2GBI.

All these experimental conditions allow the voltage sensor to experience a higher energy barrier on the deactivation pathway when compared to the activation road. To confirm this hypothesis, we performed MD simulations for *Ci*Hv1 structural modes in two distinctive active states [18,19,20]. The MD simulations suggest that an arginine at position 264 stabilizes the S4 transmembrane in an intermediate active conformation and that, consequently, the deactivation goes through an energetically unfavorable state that slows its movement.

The results presented here and our previous biophysical studies have led to important advances in our understanding of the role of the H_v_1 channel in the cells of the immunological system. Alvear et al. (2022) found that the proliferation of T lymphocytes can be modulated by inhibition of the Hv1 channel in myeloid-derived suppressor cells. In that study, possible applications in cancer therapies are discussed. Interestingly, the recent development of hydroxyl-functionalized nanoporous polymers with hollow capsular shapes can be used to deliver H_v_1 inhibitors such as zinc and 2GBI in the acidic environments of MDSCs [21].

The molecular origin of the charge trapping of the H_v_1 voltage sensor is not clear. This process could be a feature of its structure or, in principle, a feature due to the N264R mutation. The main aim of the present study is to unveil the molecular mechanism that underlies the gating charge trapping in Hv1 channels and how it relates to proton conduction and pH sensing. Our results suggest that N264R or a positive charge density within this area promotes an electrostatic repulsion on the gating charges, which facilitates an upward movement of R258 and R261. This displacement favors the formation of stable salt bridges between R261, R258 and D160 at the selectivity filter and promotes a stable configuration in the active state of the voltage sensor, favoring voltage sensor charge trapping. Here, we demonstrate that the movement of the voltage sensor is primarily influenced by hydrophobic forces in the vicinity of the selectivity filter, with electrostatic modulation occurring at position 264. Specifically, the introduction of a positive charge at position 264, whether via pH alterations, site-directed mutations or the presence of a charged molecule, results in the voltage sensor being immobilized in a trapped state. This discovery offers new insights into the interplay of permeation, voltage and pH sensing by shedding light on the phenomenon of voltage sensor trapping.

## 2. Results

### 2.1. N264R Mutation Is Crucial for Understanding the Voltage Sensor Displacement

In voltage-dependent Na^+^, K^+^ and Ca^2+^ channels, conduction and voltage sensitivity occur in different structural domains (Pore domain and Voltage-sensing domain, respectively) [22]. Unlike Shaker potassium channels, in the Hv1, the proton conduction (and selectivity), voltage and ΔpH sensitivity occur in the same structural region [10]. Highly conserved residues regulate all these processes. D160 is involved in proton selectivity and the arginines R255, R258 and R261 confer voltage-sensitivity to the Hv1 channel in *Ciona intestinalis* [9,10,11,14]. S4 helix mutations of the residues have provided molecular clues to the structure–function relationship of voltage-sensitivity in Hv1 channels. For example, the monomeric mutant channel N264R of *Ci*Hv1 produces a significant decrease in conductance (Figure 1A), which facilitates the measurement of gating currents (Figure 1B) [16,17,23]. As shown in Figure 1A, at membrane repolarization, the OFF-gating charge is much smaller than the ON-gating charge. This effect could be due to an intrinsic feature of Hv1, as previously reported for the Shaker potassium channel, in which charge immobilization, also called voltage sensor trapping, disappears when part of the N-terminus is removed [24]. Nonetheless, this is not likely to be the cause in the monomeric Hv1, since, to avoid dimerization, deletions in a large part of its C- and N-terminus are present [13].

Interestingly, when the selectivity filter is perturbed by neutralizing the aspartate 160 (ΔNΔC D160N mutant), a non-conductive mutant is obtained and the OFF-gating current amplitude is recovered when the membrane is repolarized (Figure 1B). This result suggests that the N264R mutation is probably affecting the properties of the voltage sensor. To test this hypothesis, we reintroduced the N264R mutation into the background of the ΔNΔC D160N mutant. In this double mutant, the OFF-gating current is slightly less than the ON-gating current. However, the gating charge trapping is not as pronounced when compared to that of the ΔNΔC N264R mutant (cf. Figure 1A,C). The critical observation here is that the mutation in the selectivity filter does not induce trapping. Trapping develops only when an arginine replaces the asparagine in position 264. Next, we fitted the Q(V) data for the different mutants (Figure 1D) using a two-state Boltzmann model:(1)$$\frac{Q}{Q_{max}}=\frac{1}{1+e^{\frac{z\delta F(V-V_{0})}{RT}}}$$
where zδ is the fractional displacement of the gating charges, F is the Faraday constant, V is the voltage applied and V_0.5_ is the activation energy of the voltage sensor. The comparison of the Q-V relationships from the ΔNΔC D160N mutant and double mutant ΔNΔC D160N-N264R shows minimal differences in their Boltzmann activation parameters. These results suggest that the addition of the extra arginine does not interfere with the sensor activation energy and voltage sensing. Therefore, the position of the N264R mutation in the voltage sensor activation does not reach the membrane core and, consequently, is not sensing the electric field or participating in the voltage sensing. In Figure 1E, we fitted the ON-gating current decay for ΔNΔC D160N and ΔNΔC D160N N264R mutants following the protocol proposed previously [15]. The decay kinetics seem to follow the same behavior when comparing gating currents at the ΔpH = 0 but with different intra- and extra-cellular pH. It is even possible to appreciate that at very high potentials, the decay kinetics of the ΔNΔC D160N N264R tend to be faster compared with those of ΔNΔC D160N, which give it a slightly more pronounced shape in the last voltage pulses. Based on these experiments, we hypothesize that the N264R mutation is necessary for the voltage sensor to become trapped and modify the ON-gating current decay kinetics between ΔNΔC D160N and ΔNΔC D160N N264R. Furthermore, the selectivity filter and position of N264 can interact directly; therefore, the shape of the gating currents may change. We have used variable depolarization protocols to understand these results in more detail, which are detailed below.

In Figure 2A–C, we present experiments which now utilize a time-varying depolarization protocol to explore the evolution of the temporal decay of the gating charge and gating currents for mutants ΔNΔC N264R, ΔNΔC D160N and the ΔNΔC D160N N264R double mutant. The ΔNΔC N264R exhibits a rapid reduction in the OFF/ON ratio of the gating charge (Figure 2E, yellow line and inset). The ΔNΔC D160N mutant shows a slight increase in a fast-transient OFF-gating current amplitude during approximately the first 12 ms, which then remains stable over time (Figure 2B,D). To corroborate and depict the latter, we fitted a single exponential function in the pulse protocol starting at 17 milliseconds; during this period, the amplitude of the fast OFF-gating current component remained constant (Figure 2D, purple). On the other hand, the ΔNΔC D160N N264R mutant in the same depolarization period experiences a decrease in the amplitude of the OFF-gating current (Figure 2C). When the data are fitted to a double exponential, a slow time constant that decays over time can be observed. (Figure 2D, green). This indicates that part of the fast OFF-gating current component in this rapid transition is decreasing in time.

Moreover, when we plot the ratio between the ON-gating charge and OFF-gating charge for the ΔNΔC D160N mutant (Figure 2E, purple), it is essentially constant over time, satisfying the charge conservation criteria (Q_ON_ = Q_OFF_) and indicating that the displaced charge in the ON is recovered in the OFF phase. In the case of the ΔNΔC D160N N264R mutant, the Q_OFF_/Q_ON_ ratio slowly decreases over time (Figure 2E, green). These observations indicate that, for the ΔNΔC D160N mutant, the charge transported over time is the same and that for the double mutant, the voltage sensor is slightly trapped. These findings demonstrate that arginine at position 264 is essential for maintaining the voltage sensor in an active state. Moreover, when the selectivity filter (D160) is mutated, the impact is less pronounced than in the ΔNΔC N264R mutation. The arginine in position 264 alters the H_v_1 gating properties, leading to a change in the conductance and the promotion of voltage sensor trapping.

### 2.2. Positively Charged Molecules near N264 Position Promote Changes in Voltage Sensor Displacement and Charge Trapping

To understand the mechanism of voltage sensor trapping, we introduced lysine (K) instead of arginine at position 264. These two positively charged residues differ in their size and hydrophobicity, as K has an additional methyl group in its side chain and no guanidinium group. Unlike the monomeric Hv1 ΔNΔC N264R mutant, the ΔNΔC N264K channel induces a robust macroscopic proton current and an almost negligible ON-gating current at the beginning of the depolarization stage (Appendix A). However, when the membrane is repolarized to 0 mV, corresponding to the proton reversal potential at symmetric pH, the observed OFF-gating current is like the ΔNΔC N264R current (Appendix A). Next, we introduced the N264K mutation into a ΔNΔC D160N background (Appendix A). Applying the same voltage protocol as in Appendix A to the double-mutant channel reveals an ON-gating current at the onset of depolarization accompanied by a decrease in macroscopic proton currents when compared to the ΔNΔC N264K variant. Underscoring the importance of the positive charge in position 264, Appendix A shows a considerable amount of gating charge trapping. To test this hypothesis, a key experiment would be to add a molecule with similar characteristics to arginine to the bath solution.

For that purpose, we utilized 2GBI, a classical Hv1 inhibitor [25]. This compound resembles arginine as it contains a guanidinium group. 2GBI acts with an efficiency of about 80% at a concentration of 200 μM on the intracellular side [19]. In the open state of the channel, this ligand directly interacts with the voltage sensor and the selectivity filter. The amino acids involved in its binding site, D122, F190 and R211 in hHv1, are highly conserved in Hv1 channels. In h Hv1, the binding sites are determined by an interaction where the benzo ring points toward F150 in the transmembrane segment (S2). Additionally, the imidazole ring is positioned between the selectivity filter (D112, S1) and arginine R211 (R3, S4). Moreover, R211 interacts with the guanidinium group of 2GBI [25]. When 2GBI is added to a final concentration of 1.2 mM to the recording solution on the intracellular side of the ΔNΔC D160N mutant, the blocker dramatically modifies the characteristics of the gating currents (Figure 3A,B). It modifies the ON-gating currents, akin to the ones observed for the ΔNΔC D160N N264R mutant, suggesting that arginine at this position is a key element, but not an indispensable one, in voltage sensor trapping. This is because the proposed 2GBI binding site is within the selectivity filter. It has been suggested that D160 directly interacts with the imidazole ring [18,19], stabilizing the voltage sensor via a hydrophobic interaction. In this way, 2GBI induces a voltage sensor trapping phenomenon like that produced by the ΔNΔC N264R variant. The decrease in the OFF-gating charge and the gating kinetics promoted by the inhibitor show the same pattern as the trapping induced by the ΔNΔC N264R mutant.

At this point, we observe that the 2GBI and the N264R mutation in the background ΔNΔC D160N do not seem to affect the resting-active equilibrium of the voltage sensor (Figure 3C). In the presence of 2GBI, the Q(V) data are well fitted with a V_0.5_ = 127.3 ± 4.8 mV and zδ = 1.15, in line with the Q(V) data obtained in the absence of the inhibitor.

On the other hand, the time-varying depolarization protocol (Figure 4A,B) shows the relation between ON- and OFF-gating current components. The fast OFF-gating current component rapidly decays in the presence of 2GBI when compared to the ΔNΔC D160N N264R channel (Figure 4B,C). Even when observing the Q_OFF_/Q_ON_ ratio, a function with two exponentials is needed to describe its behavior. Indeed, a function with two exponentials is needed to describe its behavior, displaying a faster decay of the OFF-gating charge over time. Thus, the interaction between mutations in D160N with the hydrophobic ring of the 2GBI allows the sensor to be trapped more efficiently. Consequently, the selectivity filter contributes to sensor movement.

### 2.3. The Hydrophobicity in the Selectivity Filter Is a Key Component in Voltage Sensor Displacement and Voltage Sensor Trapping

Even though the ΔNΔC D160N N264R and ΔNΔC D160N + 2GBI present similar behaviors in the ON-gating current kinetics, there are differences in the OFF-gating current kinetics, as evidenced in the time series of the Q_OFF_/Q_ON_ ratio over time. (Figure 2C and Figure 4B). Given this, it is necessary to systematically explore the effects of modifications within the selectivity filter. For example, the ΔNΔC D160N mutant renders robust gating currents [15]. In contrast, the D160E mutant gives rise to a macroscopic proton current (Figure 5A). On the other hand, when the D160 position is replaced by glutamine (Q), alanine (A), valine (V), cysteine (C) or isoleucine (I) (Figure 5A), different patterns in the ON-gating currents were observed as the hydrophobicity index for each residue increased [26]. Moreover, the Q(V) curves exhibit a rightward shift trend as the hydrophobicity of the amino acids augments (Figure 5B). Moreover, the slopes of the Q(V) curves (zδ values) decrease as the hydrophobicity rises. By obtaining the values of zδ and V_0.5_ (Appendix A and Equation (1)), we calculated the free energy difference (ΔΔG_mutant_) required for each mutant to move from a resting to an active state as the reference values. This ΔΔG_mutant_ energy correlates quite well with the free energy associated with the amino acid hydrophobicity (ΔG_hydrophobicity_) when we utilize the ΔNΔC D160N mutant as a reference value (Figure 5C). This means that the voltage sensor requires more energy to reach an active state as the hydrophobicity increases in the selectivity filter region. This may be a consequence of the fact that hydrophobicity can modify the transitions through which the voltage sensor is passing, affecting the activation energy of the channel as well as the displaced charge.

In summary, hydrophobicity and excess positive charges on the intracellular side near position 264 are key elements in the modulation of voltage sensor activation. Likewise, the equilibrium between the voltage sensor resting and active states is altered by the pH difference between the intracellular and extracellular sides, i.e., ΔpH [15]. Because of the high degree of coupling between ΔpH and the voltage sensor, it becomes interesting to explore whether changes in pH alter the properties of the activation and possibly the voltage sensor trapping.

### 2.4. Intracellular pH Also Promotes Voltage Sensor Trapping

As previously reported, changes in ΔpH modulate the behavior of the gating currents arising from H_v_1 channels [15]. The greater the ΔpH difference (pHo > pHi), the more the Q(V) curve is shifted leftward. Furthermore, the ON-gating current activation kinetics become faster at higher ΔpH [15]. It is also important to point out that not only is the ΔpH important, but also the absolute value of the internal pH, since higher ΔpH (more acidic on the intracellular side) increases the gating current kinetics [15], while the condition (pHo < pHi) makes the activation of the voltage sensor more difficult.

A closer examination of the time course of the currents carried by the ΔNΔC D160N mutant reveals that a large part of the ON-gating current is completed during the first milliseconds, followed by a small macroscopic current (Figure 6A, top inset). This observation suggests that the ΔNΔC D160N conducts protons, albeit with a very small efficiency. Additionally, a reduction in the OFF-gating charge is present (Figure 6A, bottom inset). When the ΔNΔC D160N N264R mutant is recorded under the same conditions, the gating current activation kinetics exhibit the same tendency (Figure 6B) but the small macroscopic current observed in the single mutant at ΔpH 2 is absent (Figure 6B, bottom inset). In this case, the OFF-gating charge is reduced but sensor trapping is less drastic.

Q(V) relationships were calculated for the ΔNΔC D160N and ΔNΔC D160N N264R mutants with a ΔpH = 2 using pHi equal to five and pHo equal to 7, and data were fitted to a two-state Boltzmann equation (Figure 6C). The results show that the Q(V) curve does not show substantial changes in activation energy between the two mutants. Likewise, the decay of the time constants calculated in the ON-gating currents also shows a similar behavior, indicating that the ON-gating charge movement develops similarly when comparing ΔNΔC D160N and ΔNΔC D160N N264R mutants (Figure 6D).

Under non-equilibrium conditions, the development of the amplitude in the fast component of the OFF-gating current changes dramatically (Figure 7A). This is because the maximum amplitude of the OFF-gating current decreases very rapidly in the first 5 ms of the time-varying depolarization protocol (Figure 7B), while the ratio of OFF- and ON-gating charges also decreases very fast, closer to 20%, with respect to the ON-gating current (Figure 7C).

Given the selectivity of the channel towards protons, increasing the acidity of the intracellular medium seems to be equivalent to locally adding positive charges within the cytosolic side of the channel. In other words, it changes the local electrostatic profile close to the N264 position. In both mutants (the ΔNΔC D160N N264R and the ΔNΔC D160N), the amplitude in the OFF-gating current is much smaller than that in the ON-gating current. This effect is promoted by acidifying the intracellular environment, since positive charges are being added on this side which force the voltage sensor to be trapped. Therefore, trapping can also be elicited by acidifying the inner vestibule of the channel near position 264, which is probably due to a transition in the kinetic model of voltage sensing highly dependent on intracellular pH. Our electrophysiological recordings point towards the idea that the hydrophobicity at the D160 position controls the voltage sensor activation. On the other hand, the electrostatic profile in the intracellular vestibule participates in trapping the voltage sensor in its active configuration. Consequently, in the following section, we took advantage of molecular modeling to explore the molecular nature of the gating charge trapping induced by the N264R mutant, which exhibited the largest degree of voltage sensor trapping (Figure 1A).

### 2.5. The ΔNΔC N264R Mutant Induces Stable Salt Bridges with the Selectivity Filter (D160) for an (Intermediate) Active-State Model

Our electrophysiological results strongly suggest that the voltage sensor activation trapping is mainly due to the N264R mutation and an interaction with D160. To explore this idea at the molecular level, we constructed two sets of active-state models of the ΔNΔC H_v_1 and ΔNΔC N264R structures based on two reported active models for hHv1. These are the Intermediate Active (A_I_) [20] and Full Active (A_F_) [18] models, which exhibit a lower and more upward displacement of the S4 helix, respectively (Appendix A). The A_I_ ΔNΔC N264R variant forms stable salt bridges (Figure 8A) over time between D160 and arginines R258-R261 (2nd and 3rd charges of voltage sensor, respectively) when compared to the A_I_ WT (Figure 8B). The distributions of the D160-R258 and D160-R261 distances over three independent simulations clearly show lower values for the N264R mutant (Figure 8A). Moreover, the time series of the three replicates for the ΔNΔC N624R show almost no fluctuations while the ΔNΔC explores a wider distance space (Figure 8A). We conclude that a positive charge placed at position 264 displaces both arginines R258 and R261 upwards and towards D160 (Figure 8C). We corroborated these findings by computing the average relative density of R255, R258 and R261 along the axis perpendicular to the membrane plane (*z*-axis) for the NΔC, ΔNΔC N264R and ΔNΔC D160N A_I_ models (Appendix A). Accordingly, relative densities for arginine in the ΔNΔC N264R variant are more displaced towards the extracellular side when compared to those of the ΔNΔC and the ΔNΔC D160 mutant.

Interestingly, when 2GBI is docked within the intracellular side of the WT A_I_ model, it locates exactly at the 258 position with its guanidinium group pointing towards D160 (Appendix A), showing similar behavior to the full active model [18]. Our models agree with the experimental data as the strong electrostatic interactions within the D160-R261-R258 triad should promote the voltage sensor trapping in the active state upon repolarization. Indeed, the fact that for the double mutant ΔNΔC D160N N264R the effects on OFF-gating currents are less severe (Figure 1C) also supports the notion that a coupling between D160 and the displaced R261-R258, triggered by arginine at position 264, are the main molecular determinants of the trapping of the OFF-gating charge experienced by ΔNΔC N624R. Remarkably, for the A_F_ models, there is no such effect. The greater upward displacement of the S4, when compared to the A_I_ model, only allows interaction between the D160-R261 pair for both the ΔNΔC and the ΔNΔC N264R models (Appendix A). In both cases, these interactions are dynamic and more stable than for the WT. Our modeling suggests that stabilizing interactions between D160 and the S4 arginines, for the case of the ΔNΔC N264R mutant, are distinctive when compared to the WT only for the A_I_ model.

To translate these findings into a more intuitive and numerical value, we estimated the electrostatic interaction energy present in the R258-R261-D160 triad via a simplistic Coulombic energy calculation, as performed elsewhere [27]. Briefly, from the time series of the distances between the center of mass of the charged groups from interacting pairs (D160-R258, D160-R261 and R261-R258), we computed average Coulomb energies and, by adding these three terms, we estimated the energy differences of the A_F_ ⟶ A_I_ transition for the ΔNΔC and ΔNΔC N264R variants which are presented in Appendix A (2nd to 5th columns). As expected, for the ΔNΔC N264R mutant, this transition has an energy cost of around 3 kcal/mol which is due to the formation of the two salt bridges in the A_I_ state. On the other hand, the WT exhibits an A_F_ ⟶ A_I_ transition of around 0.4 kcal/mol, within thermal noise and being a consequence of the salt-bridge exchange without the formation of an extra one (6th column of Appendix A). Consequently, in the context of a thermodynamics cycle, the difference between the A_F_ ⟶ A_I_ transitions for the ΔNΔC N264R and ΔNΔC models is estimated to be about 2.6 kcal/mol (7th column of Appendix A), reflecting the more rugged shape of the potential energy surface of the ΔNΔC N264R variant.

In Appendix A, the model that best reproduces the gating current energetics of the ΔNΔC N264R variant (see Appendix A for model parameters) suggests the existence of at least two distinctive active states, while the ΔNΔC D160N and the ΔNΔC D160N N264R mutants effectively modify the resting barrier states; in other words, there is a coupling between the D160 and arginine at position N264. The fact that the kinetic model of the ΔNΔC N264R mutant reveals the existence of two active states is in line with the different salt-bridge pattern between D160 and S4 arginines observed for the A_I_ and A_F_ models.

## 3. Discussion

The monomeric *Ci*H_v_1 is a relevant system for studying the H_v_1 voltage sensor’s properties and how they relate to the functions of the channel and its structure. The monomeric (ΔNΔC) *Ci*Hv1 retains most of the characteristics reported for the wild-type dimer Hv1, such as high proton selectivity, sensitivity to ΔpH, voltage dependence and inhibition by 2GBI. These processes are related to sections of the protein that are highly conserved. For example, the ΔNΔC D160N mutant almost completely isolates the proton current, allowing the voltage sensor movement to be detected. It is also known that changes in the ΔpH of the channel modify the active–resting equilibrium of the voltage sensor depending on the difference between the external and internal pH, thus also involving the arginines that make up the voltage sensing.

Additionally, ΔpH adjustments modify the kinetics and activation energy in Hv1 voltage sensor. Moreover, 2GBI provides insights into the spatial distribution of amino acids and the distances between critical elements such as the selectivity filter, certain arginines comprising the voltage sensor and phenylalanine related to the gating charge transfer centre; in addition, 2GBI effectively prevents the passage of protons. Also, the N264 position is crucial in understanding proton passage through H_v_1 channels. Accessibility experiments with MTS reagents demonstrate that their binding to cysteine at this site significantly decreases current density [12,28]. This suggests this position strongly influences the channel permeation pathway while preserving proton selectivity [12,29]. Furthermore, these experiments demonstrate that the N264 residue in the S4 transmembrane segment (voltage sensor) can bind to the MTS reagents when exposed to the solvent in the open state. Substituting it with an arginine also leads to decreased proton conduction and activation kinetics, slow tail currents, and a leftward shift in activation energy compared to the wild type [16,17]. Considering the information above, the ΔNΔC N264R mutation enables the recording of gating currents due to its fast activation kinetics and decreased proton current. However, two effects emerge: first, the behavior of the voltage sensor in the ON-gating current overlaps with the proton current, and second, there is a decrease in the OFF-gating charge upon returning to the proton reversal potential. The OFF-gating currents show only a component carrying a small fraction of the total charge observed in the ON-gating current component. This suggests that the gating particles must surmount a significant energy barrier to return to their resting state and point towards a charge-trapping state of the voltage sensor due to the N264R mutation. Therefore, N264R can be directly related to the permeation of protons and the movement of the voltage sensor when it is activated.

Altering the Hv1 selectivity filter by mutating the aspartate 160 to an asparagine produces a nonconductive channel in which the ON-gating charge is recovered during membrane repolarization [15] Moreover, in the ΔNΔC D160N N264R, the activation kinetics of the ON-gating currents render similar effects when compared to the ones produced by the ΔpH changes [15]. Our 5-state gating kinetic model (Appendix A) indicates that the difference in trapping between the mutants ΔNΔC D160N-N264R and ΔNΔC N264R is due to differences in the energy landscape to governs the movement of the gating charges in both cases. It is apparent from Appendix A, that the rate limiting step is in the case of ΔNΔC N264R is considerably larger that for the double mutant. Additionally, an arginine at position 264 does not alter the voltage sensor resting–active equilibrium since the V_0.5_ and zδ for both ΔNΔC D160N and ΔNΔC D160N N264R, respectively, are essentially equivalent. In other words, arginine at position 264 is not part of the Hv1 voltage-sensing process. Our findings indicate that reintroducing the N264R mutation in the background of the ΔNΔC D160N results in minimal trapping of the voltage sensor. We suspect that there is an interaction between the selectivity filter (D160) and the N264R, leading to reduced charge trapping. This is possibly due to the absence of any modulating component, such as hydrophobicity in the proximity of the selectivity filter, in the ΔNΔC D160N.

Previous reports have shown that modifications in the selectivity filter in hHv1, such as D112V, render non-conducting channels. Other mutations, such as D112Q, D112N and D112A, produce very small currents. In the wild-type channel, *Ci*Hv1 mutations such as D160N, D160A and D160C prevent proton conduction. When these mutations occur in the monomeric channel, gating currents in the absence of proton currents can be recorded. In contrast, the expression of the ΔNΔC D160E mutant produces robust proton currents.

The hydrophobicity index of each amino acid also modulates the voltage sensor movement. As the hydrophobicity increases, the V_0.5_ of the Q(V) curve is shifted rightward, increasing the activation Gibbs free energy of the voltage sensor. These rightward shifts may result from the residue D160 being close to two hydrophobic regions above and below the selectivity filter [10,30,31]. Increasing the hydrophobicity of residue 160 could increase the strength of interaction between the residue 160 and the hydrophobic regions, hindering the voltage sensor displacement.

Interestingly, not only a positive charge at position 264 is important in the voltage sensor movement. When this position is replaced by a lysine, changes at the repolarization are observed and a large macroscopic current appears, followed by a small OFF-gating current. If the D160N mutation is added to the latter, it exhibits a considerable decrease in the macroscopic current and the trapping of the OFF-gating current, supporting the idea that the physicochemical nature of an arginine at position 264 and its charge modulate the voltage sensor. 2GBI, a known intracellular Hv1 inhibitor that resembles arginine, also produces gating charge trapping in the ΔNΔC D160N mutant, an effect that resembles the one observed for the ΔNΔC N264R variant.

About 60% of the chemical potential stored in the ΔpH is used by the voltage sensor to activate the channel [15]. As the pH and voltage-sensing processes are intimately related, changes in pH can be attributed to local changes in positive charges around the channel permeation pathway and close to position 264. Therefore, it is expected that the pH can also modulate the voltage sensor movement. Our results show that the more acidic the pH is on the intracellular side, the faster the activation kinetics become [15], while the OFF-gating current is also affected, showing a decrease in the transported OFF-gating charge, i.e., a low internal pH promotes voltage sensor trapping. The effect of pH on the OFF-gating current is probably due to the local presence of positive charges close to position 264. In this line, a small and positively charged inhibitor may provide insights into this mechanism. The small and positively charged ligand called 3-(2-amino-5-methyl-1H-imidazol-4-yl)-1-(3,5-difluorophenyl) propan-1-one or H_v_1 Inhibitor Flexibles (HIF) is more efficient than 2GBI [32,33]. This molecule, due to its size and flexibility, can access a very narrow region by interacting with charged residues and also with the voltage sensor by blocking the opening of the channel on the intracellular side via the protonation of the 2-aminoimidazole ring group on one of its mobile rings [32].

Our results show that 2GBI, ΔpH and arginine share characteristics that allow voltage sensor trapping in a similar manner to the ΔNΔC N264R mutant. To explore this notion at a molecular level, we carried out MD simulations of two active-state ΔNΔC *Ci*H_v_1 models with different upwards displacements of the S4 helix. Molecular models suggest that the hydrophobic plug present in the Hv1 closed state is disrupted when the channel is activated and the S4 alpha helix moves upwards [30]. Our MD simulations show that it is possible to trap the voltage sensor only for an intermediate state (A_I_ model). The A_I_ model proposes that trapping is the result of the formation of long-lasting salt bridges between D160 and the R258 and R261 of the ΔNΔC N264R mutant (Figure 8). A model that exhibits a further upward displacement of the S4 helix reveals no noticeable difference in salt-bridge interactions between the D160 and S4 helix when compared to the WT. The formation of stable salt bridges in the A_I_ model reveals that the biophysical profile of the ΔNΔC N264R mutant is a consequence of the presence of the arginine at position 264, which electrostatically repels R258 and R261, stabilizing this intermediate active configuration of the voltage sensor. In a similar fashion, the 2GBI guanidinium group interacts with the arginines of the voltage sensor, allowing a stable configuration of the active state of the voltage sensor. Moreover, docking studies of 2GBI (Appendix A) revealed that the guanidine group of 2GBI interacted with the selectivity filter and pointed towards the second arginine (see also [18]).

The five-state kinetic model of ΔNΔC N264R ([16]; Appendix A) is able to account for the characteristics of the mutant gating currents presented here. Minor modifications to the parameters of the kinetic model allow the qualitative representation of the gating currents. This model also indicates that the motion of the voltage sensor, in general, occurs mainly in two steps. In the case of the N264R mutant, upon depolarization, there is a fast-gating charge movement between states A3 and B1. Still, once the membrane is repolarized, the charges must surmount a significant energy barrier between B1 and A3 (Appendix A). Therefore, the charge is trapped in the energy well defined by B1. In the case of mutants ΔNΔC D160N + ΔpH > 0 and ΔNΔC D160N N264R, ΔNΔC D160N + 2GBI decreasing this energy barrier accounts for the degree of trapping in each of these experimental conditions (Appendix A). The predictive strength of the model suggests that it is a general kinetic mechanism for the *Ci*Hv1 voltage sensor movement. To break a salt bridge, a range between 1–10 kJ/mol [34] is necessary, which will be equivalent to breaking the D160-R258/D160-R261 (in ΔNΔC N264R) or ΔNΔC D160N + 2GBI interactions. To find out how long it takes to bring H_v_1 out of this trapping state, it is necessary to use a voltage protocol consisting of two time-varying depolarizations in the ΔNΔC D160N mutant with 2GBI (Figure 9A). By fitting an exponential function (Figure 9B), it is possible to determine a time constant of 35.6 ± 0.2 ms, which is a very long time for the voltage sensor to overcome this kind of “friction” to reach its relaxed state.

Finally, this study on voltage sensor trapping offers the opportunity to learn in more detail how the processes of ΔpH and voltage sensing operate and how the channel permeation pathway functions in an orchestrated way in the Hv1 active state while allowing a new understanding of the structure–function relationship of the Hv1 channel.

## 4. Materials and Methods

### 4.1. Sequencing, Site-Directed Mutagenesis and Transcription

The *Ciona intestinalis* H_v_1 Hv1 clone was provided by Dr. Yasushi Okamura. The clone is contained in a pSP64T vector which was modified at E129 with a methionine start codon and V270 with a stop codon to produce a monomeric version of the channel (ΔNΔC *Ci*Hv1).

The mutations at the D160 and N264 positions were inserted using a QuickChange kit (Agilent, Santa Clara, CA, USA). Subsequently, PCR amplifications of DNA were checked via sequencing and later digested using the restriction enzyme PvuII (Thermoscientific, Waltham, MA, USA). The transcription of the linearized DNA was carried out using a mMESSAGE mMACHINE SP6 transcription kit (Ambion, Austin, TX, USA). Finally, the RNA was quantified by measuring the ultraviolet absorbance at 260 nm and its quality was controlled using electrophoresis in agarose gel.

### 4.2. Oocyte Extraction Procedure and RNA Microinjection

The protocol used to manipulate and extract the *Xenopus laevis* oocytes was previously described [15,16]. Oocytes were injected at a concentration of 1 µg/µL in 50 nL of RNA, were later incubated at 18 °C and finally measured between 1 and 3 days.

### 4.3. Electrophysiology

All the electrophysiological data were obtained using an Axopatch 200B amplifier (Axon Instruments, Union City, CA, USA). The voltage and current were analogically filtered using 8-pole Bessel low-pass filters (Frequency Devices, Ottawa, IL, USA) through a 16-bit analog-to-digital converter (Digidata 1440A, Axon Instruments) with a sampling frequency of 250 kHz. The data were extracted using Clampex 10.6 and then processed using scripts created in Python. The linear component of the capacity current was subtracted analogically using the analog-to-digital converter and additionally subtracted using the -P/8 protocol.

The methodology used to measure currents was the patch-clamp technique in the inside-out mode in *Xenopus laevis* oocyte membranes. The pipettes were made using borosilicate capillary glass (1B150F-4, World Precision Instruments, Sarasota, FL, USA) and subsequently stretched using a horizontal puller (Sutter Instruments, Novato, CA, USA) and fire-polished (Narishige, London, UK) to achieve diameters of approximately 15 µm to 20 µm (giant patches).

The intracellular and extracellular bath solutions contained 100 mM HEPES buffer, 50 mM N-methyl-D-glucamine (NMDG)-methanesulfonate and N-methyl-D-glucamine (NMDG)-methanesulfonate adjusting solutions to pH 5.0 and 7.0, 1 mM of EGTA (Ethylene glycol-bis(2-aminoethylether)-N,N,N′,N′-tetraacetic acid) and 2 mM MgCl_2_. All experiments were performed at a controlled room temperature of 22 °C, and only one Q(V) protocol was measured per patch to avoid changing the properties of the currents.

### 4.4. Kinetic Modeling

The gating currents were simulated by solving a system of first-order differential equations of the form dF(t)/dt = N F(t) Z, where N is the total number of channels, F(t) is a matrix containing the time-dependent probability of each state and Z is the vector containing the gating charge associated to each state. The matrix differential equation was solved by finding the eigenvalues and eigenvectors of the associated characteristic equation. The voltage-dependent kinetics constants were assumed as α(V) = α_0_ exp(xzδe_0_V/k T) and β(V) = β_0_ exp((x − 1) zδe_0_V/k T), where xzδ e_0_ and (x − 1)zδe_0_ is the fraction of gating charge moved in each transition (forward and reverse direction, respectively), k is the Boltzmann constant and T is the temperature.

### 4.5. Molecular Modeling

The *Ci*H_v_1 structure (monomer, residues 136 to 269) was generated via comparative modeling by satisfaction of spatial restraints using Modeller v9.22 [35] utilizing the H_v_1 mouse chimera structure, PDB code (3WKV) [10]. Missing residues (122–131, 159–162 and 187–190 from template) were included via loop modeling. The 3WKV structure has been proposed to be in an Intermediate Resting state (IR). Thus, two reported Active models, the A_I_ and A_F_ models, which exhibit a lower and a higher degree of upward displacement of the S4 segment, respectively, were generated as well [20]. The A_I_ state was generated as follows: the S4 helix was displaced towards the extracellular side by approximately 0.5 nm. This was achieved by moving the S4 helix three positions towards the N-terminal in the sequence alignment to build comparative models. The A_F_ model was obtained by employing as the template an active hH_v_1 structure which was the result of extensive microsecond MD under depolarizing potentials [18]. For a depiction of both models, please refer to Appendix A.

Molecular dynamics (MD) simulations were performed using the NAMDv2.13 software [36] and the CHARMM32 force field [37,38] with periodic boundary conditions and explicit solvent (TIP3P water model [39]) and a POPC membrane with a box size of 10 × 10 × 7 nm (around 70,000 atoms). Pressure (1 atm) and temperature (310 K) coupling was performed using Langevin dynamics and the Nose–Hoover method with a 1 ps-1 damping coefficient [40]. A 1.2 nm cut-off of 1.2 was applied for Lennard-Jones and real space electrostatics, with a smoothing function applied within 1.0 nm and 1.2 nm. Long-range electrostatic interactions were calculated using the Particle Mesh Ewald method [41]. The SHAKE algorithm [42] was applied to all bonds involving hydrogen atoms. We used a multiple time-step scheme utilizing the rRESPA algorithm with 2 fs for bonded and real-space interactions and 4 fs for long-range electrostatics. Na^+^ and Cl^−^ were added, reaching electro-neutrality and a final concentration of 0.15 M.

Before production runs, minimization via conjugate gradient and relaxation with MD runs applying a temperature rescaling scheme were performed until the desired temperature was obtained. For each model, 3 simulations of 350 ns were simulated, discarding the first 50 ns.

The docking procedure was performed using AutoDock Vina 1.2.0 [43]. The docking grid was defined as the intracellular portion of the channel. To obtain a representative structure of the most probable docking pose, we initially clustered the ensemble of structures from the 3 MD replicas via a k-means algorithm with the RMSD (Cα) as the metric; the docking procedures were carried out utilizing the centroid of the most populous cluster.

## 5. Conclusions

Since all the amino acids involved in this work are highly conserved in all the Hv1 species identified, charge trapping has allowed us to understand the molecular workings of the H_v_1 voltage sensor further. The following points underscore our findings: (i) The arginine in the mutant R264N does not contribute to the Hv1 voltage sensitivity, i.e., it is outside the electric field. (ii) The hydrophobic index in the channel selectivity filter also functions as an energy barrier that modulates the active state of the voltage sensor. (iii) It is possible to modulate the active state of the voltage sensor by introducing positive charges in an area close to the N264 position. (iv) The five-state gating kinetic model establishes a general description of the energy landscape that determines the displacement of gating charges in the electric field. The energy landscape is modulated by positive charges around position 264 and by the degree of hydrophobicity of the selectivity filter. Thus, this work gives us a view of how the voltage sensor simultaneously relates to the processes of permeation, pH and inhibition by 2GBI via charge trapping in H_v_1 voltage-gated proton channels.

## Figures and Tables

**Figure 1 ijms-25-00426-f001:**
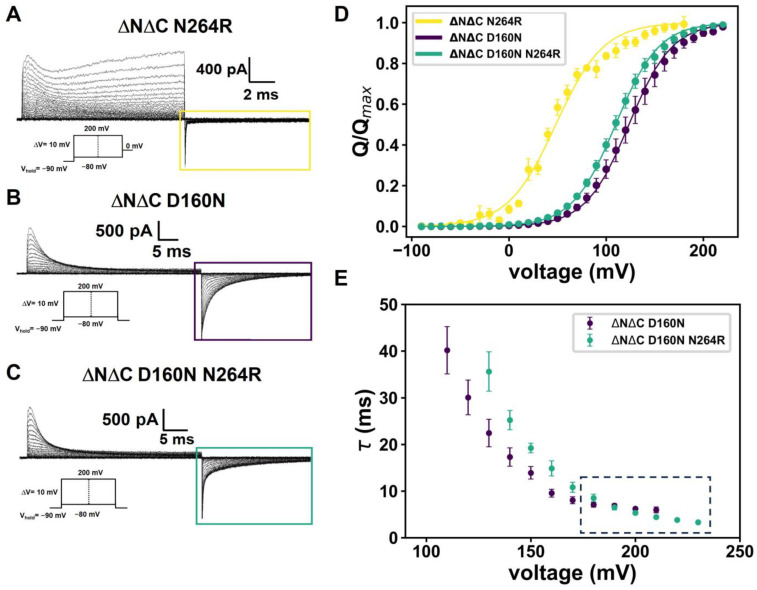
N264R promotes charge trapping. Representative gating currents for (**A**) ΔNΔC N264R, (**B**) ΔNΔC D160N and (**C**) ΔNΔC D160N N264R. Colored rectangles represent modifications in OFF-gating currents produced by the arginine in the 264 position. (**D**) Q(V) relation was fitted using two-state Boltzmann equation. The values obtained were V_0.5_ = 48.6 ± 1.2 mV and zδ =1.04 ± 0.10 for ΔNΔC N264R (N = 4), V_0.5_ = 124.1 ± 6.2 mV and zδ =1.14 ± 0.03 for ΔNΔC D160N (N = 8) and V_0.5_ = 117.0 ± 2.2 mV and zδ = 1.18 ± 0.02 for ΔNΔC D160N N264R (N = 7). (**E**) Mean ON-gating current decay time constants of ΔNΔC D160N and ΔNΔC D160N N264R mutants as a function of voltage, fitted to a single exponential function I_g_(t) = I_0_ exp(−t/τ). Data are shown as MEAN ± SEM.

**Figure 2 ijms-25-00426-f002:**
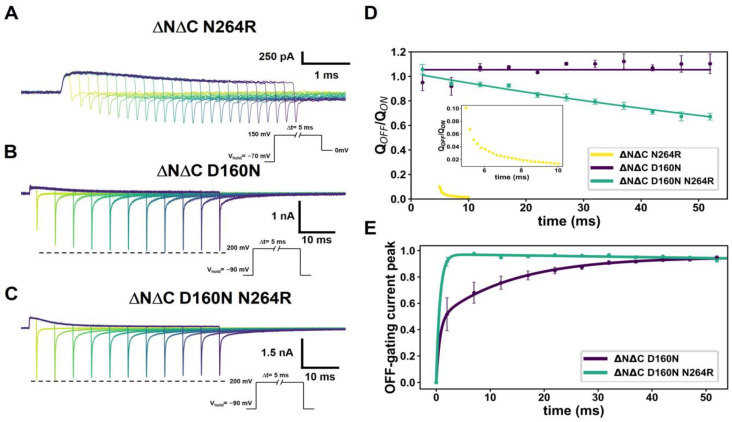
Time course of voltage sensor gating charge trapping due to N264R mutation. (**A**–**C**) represents traces of currents produced by time-varying depolarizations for ΔNΔC N264R, ΔNΔC D160N and ΔNΔC D160N N264R, respectively, with the length of the pulse is represented by darker colors. (**D**) Maximum peaks of fast transient OFF-gating currents in function of time depolarizations for ΔNΔC D160N and ΔNΔC D160N N264R. A double-exponential function was fitted, as follows: I_g_ = A (1 − exp(−t/τ_1_)) + B (1 − exp(−t/τ_2_)). The time constants calculated were τ_1_ = 0.69 ± 0.09 ms and τ_2_ = 15.66 ± 0.74 for ΔNΔC D160N (N = 4) and τ_1_ = 0.70 ± 0.06 ms and τ_2_ = 1175.13 ± 247.19 for ΔNΔC D160N N264R (N = 5). (**E**) The Q_OFF_/Q_ON_ as a function of time for ΔNΔC D160N and ΔNΔC D160N N264R mutants. A straight line near Q_OFF_/Q_ON_ = 1.05 ± 0.00 was fitted for ΔNΔC D160N and ΔNΔC D160N N264R was fitted to an exponential function (A exp((−t/τ_1_)) with parameters τ_1_ = 121.77 ± 10.32 ms. Time course of charge trapping shown as ratio of Q_off_/Q_on_ as a function of time. The ON-gating charge was calculated by integrating each depolarizing pulse of the ON-gating current at successive times of 2, 7, 12, 17, 22, 27, 32, 37, 42, 47 and 52 ms while the OFF-gating charge was calculated from the OFF-gating current during the first 20 ms. Data shown as MEAN ± SEM.

**Figure 3 ijms-25-00426-f003:**
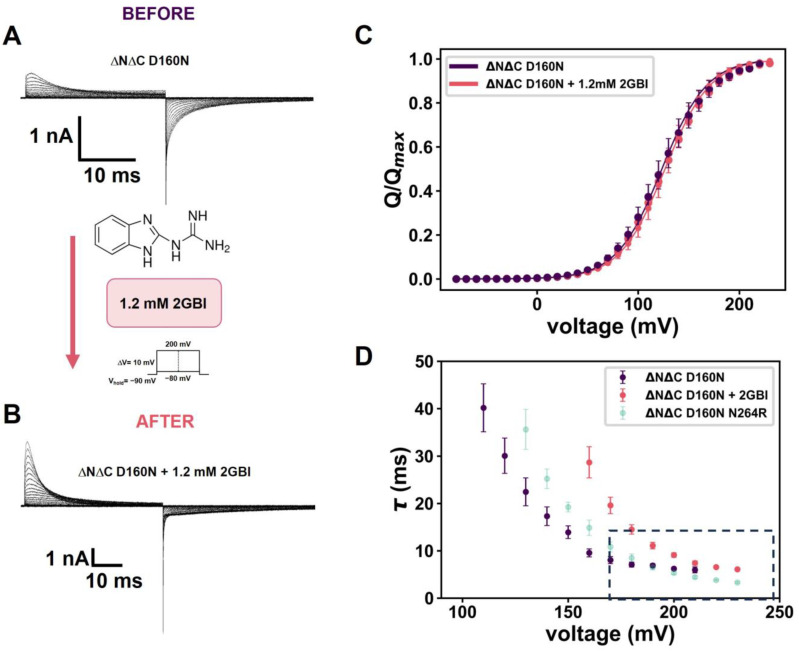
2GBI mimics ΔNΔC N264R OFF-gating current. (**A**,**B**) Representative gating currents of ΔNΔC D160N before (**A**) and after (**B**) perfusion of 1.2 mM 2GBI. (**C**) Q(V) relation was fitted using two-state Boltzmann equation in ΔNΔC D160N (purple) and ΔNΔC D160N + 2GBI (pink). The parameters obtained were V_0.5_ = 127.3 ± 4.8 mV and zδ =1.18 ± 0.08 (N = 9). (**D**) ON-gating current time decay of ΔNΔC D160N mutants + 2GBI as function of voltage, fitted as a single-exponential function I_g_(t) = I_0_ exp(−t/τ). Data shown as MEAN ± SEM.

**Figure 4 ijms-25-00426-f004:**
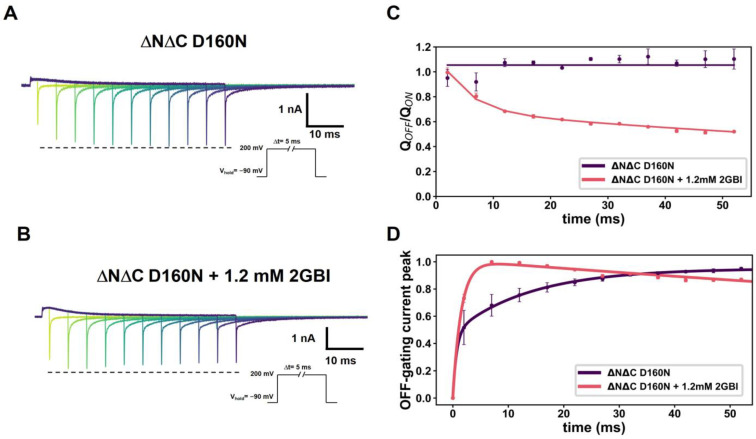
2GBI modifies properties of voltage sensor kinetics. (**A**,**B**) Representative traces of currents produced by time-varying depolarizations for ΔNΔC D160N before (**A**) and after (**B**) perfusion of 1.2 mM of 2GBI. (**C**) Maximum peaks of fast transient OFF-gating currents in function of time depolarizations for ΔNΔC D160N in presence of 2GBI perfusion (darker color-schemes represents longer depolarization intervals). A double-exponential function was fitted as follows: I_g_ = A (1 − exp(−t/τ_1_)) + B (1 − exp(−t/τ_2_)). The time constants calculated were τ_1_ = 1.51 ± 0.08 ms and τ_2_ = 333.12 ± 29.22 ms for ΔNΔC D160N (N = 5). (**D**) The Q_OFF_/Q_ON_ as a function of time for ΔNΔC D160N + 1.2 mM 2GBI was fitted to A exp(−t/τ_1_) + B exp(−t/τ_2_), and the parameters found were τ_1_ = 185.25 ± 18.13 ms and τ_2_ = 5.00 ± 0.78 ms. The depolarization time of the ON-gating current at successive times of 2, 7, 12, 17, 22, 27, 32, 37, 42, 47 and 52 ms while the OFF-gating charge was calculated from the OFF-gating current during the first 20 ms. Data shown as MEAN ± SEM.

**Figure 5 ijms-25-00426-f005:**
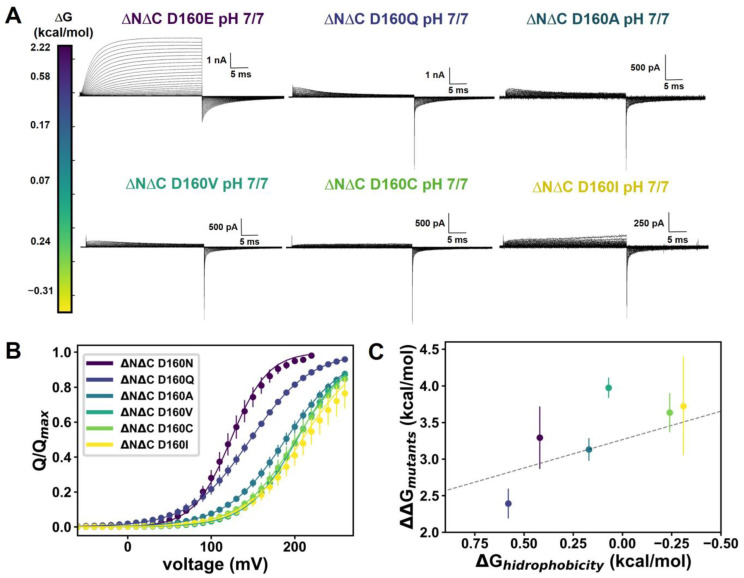
Changes in the hydrophobicity of the selectivity filter result in changes in the voltage sensor displacement. (**A**) Gating currents produced in patches expressing different mutants in D160 (selectivity filter) position. From top to bottom, the hydrophobicity index has changed; the more negative the energy value, the more hydrophobic the amino acid. (**B**) Normalized Q(V) curves for different mutations in D160, a two-state Boltzmann distribution was adjusted for each one (see fit values in Appendix A) at same symmetrical pH 7/7 (ΔpH = 0). (**C**) Linear correlation between the energy to move from an active to an inactive state of the voltage sensor and the hydrophobicity index for each amino acid. Data shown as MEAN ± S.D.

**Figure 6 ijms-25-00426-f006:**
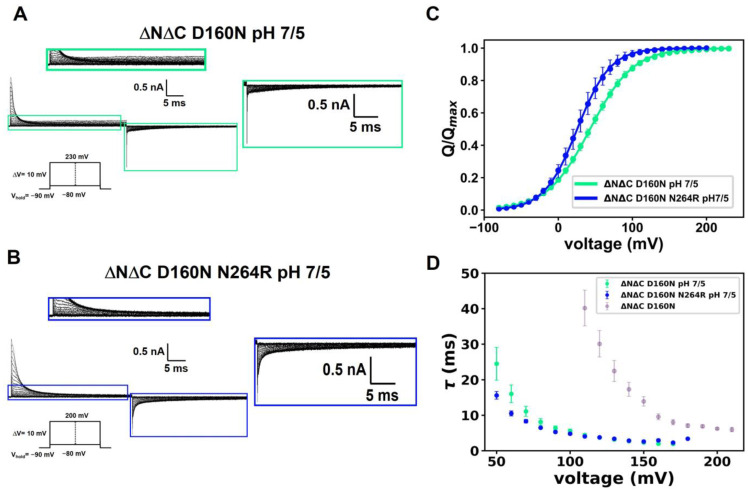
Acidity modifies OFF-gating currents. (**A**,**B**) Superimposed traces of gating currents of (**A**) ΔNΔC D160N and (**B**) ΔNΔC D160N N264R to ΔpH = 2 (pH_o_ = 7 and pH_i_ = 5). The upper inset of (**A**) shows how the ΔNΔC D160N mutant carries a small amount of macroscopic current (green) whereas, in the ΔNΔC D160N N264R mutant, this current disappears (blue). In both mutants, it can be seen that the OFF-gating current becomes smaller compared to the ON-gating current (lower insets). (**C**) Q(V) curves of ΔNΔC D160N (green) and ΔNΔC D160N N264R (blue) to pH 7/5. Curves were fitted using a two-state Boltzmann distribution. Data shown as MEAN ± S.E.M.

**Figure 7 ijms-25-00426-f007:**
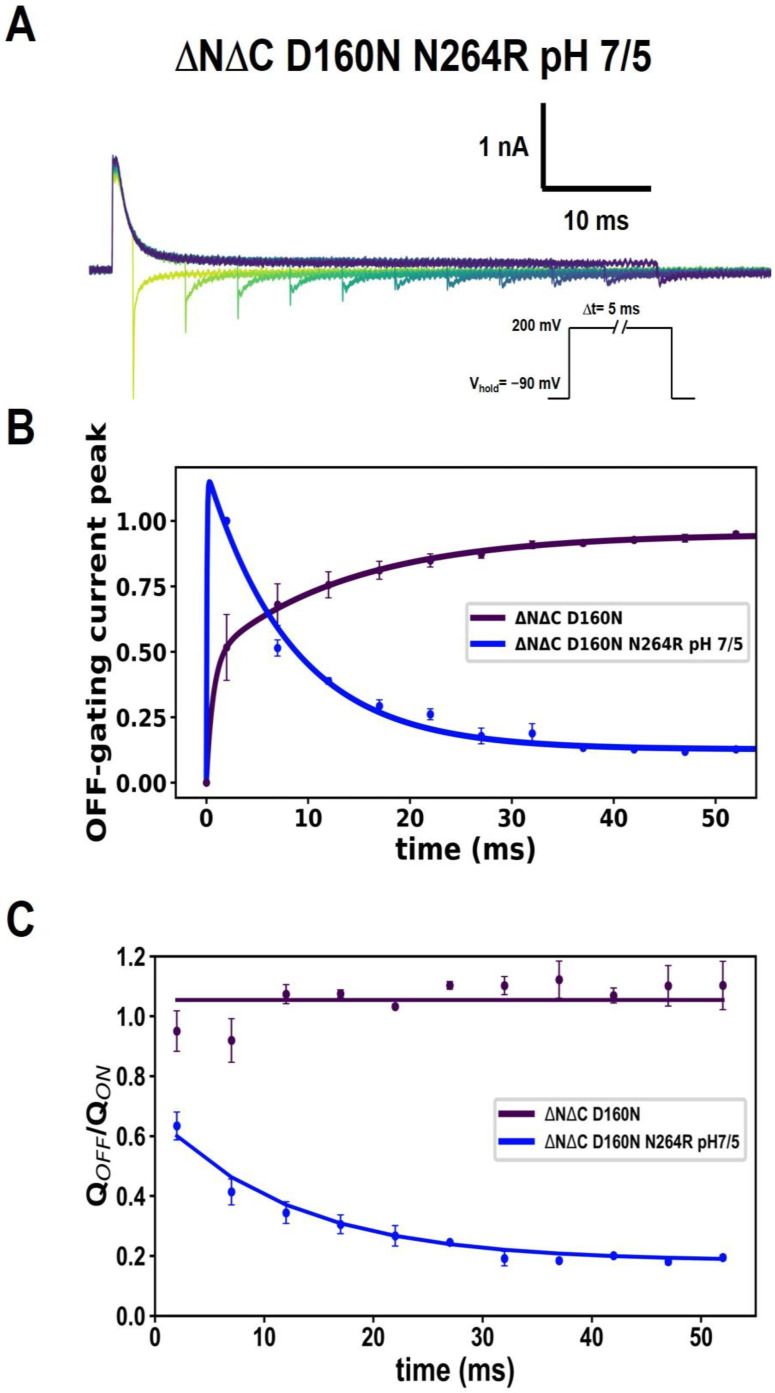
Acidity also promotes charge trapping. (**A**) Representative traces of currents (in colors) produced by time-varying depolarizations for ΔNΔC D160N N264R. The depolarization time of the ON-gating current at successive times of 2, 7, 12, 17, 22, 27, 32, 37, 42, 47 and 52 ms. (**B**) Maximum peaks of fast transient OFF-gating currents in function of time depolarizations using ΔpH = 2 (pHo = 7 and pHi = 5) for ΔNΔC D160N N264R. A double-exponential function was fitted to obtain the time constants, I_g_ = A (1 − exp(−t/τ_1_)) + B (1 − exp(−t/τ_2_)); the values found were τ_1_ = 0.0632 ± 0.00 ms and τ_2_ = 8.40 ± 0.73 ms (N = 4). (**C**) Q_OFF_/Q_ON_ time depolarization course of ΔNΔC D160N N264R ΔpH = 2, charge ratio was fitted as follow: A exp(−t/t_1_) + B exp(−t/t_2_), the parameters found were τ_1_ = 6.39 ± 1.59 ms and τ_2_ = 87.57 ± 32.90 ms. Data shown as MEAN ± SEM.

**Figure 8 ijms-25-00426-f008:**
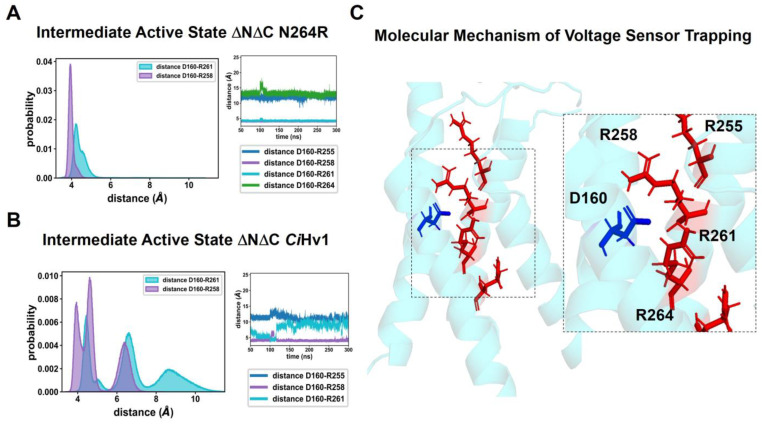
Molecular mechanism of charge trapping. (**A**) Distance probability distributions of A_I_ model of ΔNΔC *Ci*H_v_1. The right panel shows a representative time-series interaction of position D160 (blue) with arginines 258 and 261 (red). (**B**) When the N264R mutation is placed, the distance probability distributions in the A_I_ model collapse at values corresponding approximately to salt bridge interactions. The right panel shows how the time series remains, on average, much longer at values close to 4 Å. (**C**) Representative structure of a cluster of structures showing that the selectivity filter D160 and R258 and R261 are located at a position near 4 Å when the N264R mutation is placed.

**Figure 9 ijms-25-00426-f009:**
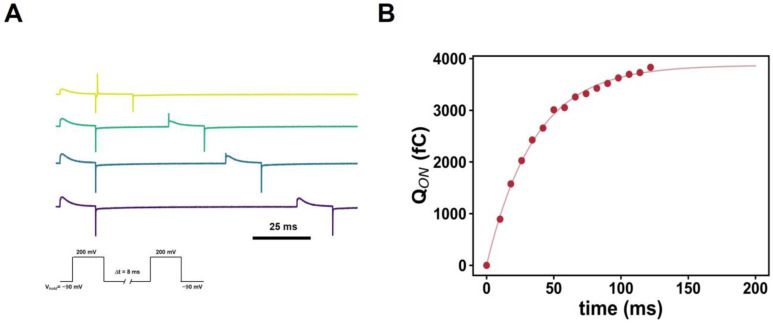
2GBI-ΔNΔC D160N interaction recovers the gating charge via a voltage pulse protocol. (**A**) ON-gating current recovery protocol. The protocol consists in two depolarizing pulses from a holding potential of −90 mV to 200 mV which were separated by an interval of time between the depolarizing pulses that increased 5 ms between traces (shown in different colors for differentiation). (**B**) The charge displaced during the second depolarization as a function of the time interval between the contiguous depolarizations. The data were fitted using a single increasing exponential function Q [exp(−t/τ)] where Q is the maximum value of amplitude 3872.52 ± 37.42 fC and τ =35.63 ± 1.1 ms is the time constant.

## Data Availability

Data is contained within the article and Appendix A.

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
