# Peer review of "Trapping Charge Mechanism in Hv1 Channels (CiHv1)"

_ijms, 2023, doi:10.3390/ijms25010426_

Round 1

Reviewer 1 Report

Comments and Suggestions for Authors

The manuscript presented a detailed and complex scientific discussion on the functional and structural aspects of the Hv1 channel, specifically focussing on the phenomenon of voltage sensor trapping. However, a few issues need to be clarified.

1.       Please explain the significance of the observation that the gating charge trapping is less pronounced in the double mutant (ΔNΔC D160N-N264R) compared to the single mutant (ΔNΔC N264R). The authors should clearly articulate the implications of this difference in the context of the research objectives.

2.       In Section 2.1 the authors discuss the decrease in the amplitude of the OFF-gating current for the ΔNΔC D160N N264R mutant and mention a slow time constant. Please, add a brief interpretation or hypothesis about the significance of this result in the context of the research.

3.       In Section 2.2 the authors discuss the introduction of the Hv1 inhibitor 2GBI. Please, provide a brief explanation of how this compound interacts with the channel and clarify the significance of the ligand and how it mimics the effects of the ΔNΔC N264R mutant.

4.       In Figure 5C, the linear correlation between the energy to move from an active to an inactive state and the hydrophobicity index is discussed. Please present a concise interpretation of the significance of this correlation in the context of voltage sensor activation.

5.       Please add a concise interpretation of the results presented in Figure 7, especially regarding the time course of ΔNΔC D160N N264R currents under ΔpH=2 conditions and explain the observed changes and their implications.

6.       In the discussion section, please consider adding more explanations of the results. For example, provide a brief explanation of why the monomeric CiHv1 is a relevant system for studying the properties of the Hv1 voltage sensor and explain why this system is suitable for exploring the relationships between the function and structure. Also, please clearly articulate the implications of the N264R mutation on gating currents. Discuss why the severe reduction in macroscopic currents and the observed behaviour of OFF-gating currents suggest charge trapping in the voltage sensor. The role of N264 position in Hv1 should be also disused in more detail. Give a brief description of the broader significance of the N264 position, not only in the movement of the voltage sensor but also in proton passage within Hv1. Discuss how alterations at this position affect current density, kinetics, and activation curve, and how these changes align with the state-dependent exposure to the solvent.

7.       In my opinion, the authors should point out the connections between the results of the study and their broader significance. In this way, it will be possible to clarify the implications of the results for understanding the function and structure of Hv1.

Author Response

Please find enclosed the revised version of our manuscript entitled “Trapping charges Mechanism in Hv1 channels (CiHv1)” by Miguel Fernández, Juan J Alvear-Arias, Emerson Carmona, Christian Carrillo, Antonio Peña-Pichicoi, Erick O. Hernández-Ochoa, Alan Neely, Osvaldo Alvarez, Ramon Latorre, Jose Garate, Carlos Gonzalez that we are resubmitting for review and publication as an article the special issue “State-of-the-Art Molecular Biology in Chile, 2nd Edition” of IJMS
The manuscript was changed according to the Reviewer 3’s comments and we have highlighted the changes in yellow on the text. We are grateful for the commitment to improve the quality of manuscript and hope. Hopefully this new version of the manuscript is suitable for the potential publication in IJMS.

Reponses to Reviewer #3:

  1. Please explain the significance of the observation that the gating charge trapping is less pronounced in the double mutant (ΔNΔC D160N-N264R) compared to the single mutant (ΔNΔC N264R). The authors should clearly articulate the implications of this difference in the context of the research objectives.

We added some sentences on the bottom of page 18:

Answer: Our 5-state gating kinetic model (Fig. S6) indicates that the difference in trapping between the mutants ΔNΔC D160N-N264R and ΔNΔC N264R is due to differences in the energy landscape to governs the movement of the gating charges in both cases. It is apparent from Fig. S6, that the rate limiting step is in the case of ΔNΔC N264R considerably larger that for the double mutant.

  1. In Section 2.1 the authors discuss the decrease in the amplitude of the OFF-gating current for the ΔNΔC D160N N264R mutant and mention a slow time constant. Please, add a brief interpretation or hypothesis about the significance of this result in the context of the research.

Answer: We added some sentences on page 7 to clarify the point raised by the reviewer: Based on these experiments, we hypothesize that the N264R mutation is necessary for the voltage sensor to become trapped and modify the ON-gating currents decay kinetics between ΔNΔC D160N and ΔNΔC D160N N264R. Furthermore, the selectivity filter and position N264 can interact directly; therefore, the shape of the gating currents may change. We used variable depolarization protocols to understand these results in more detail, which are detailed below

  1. In Section 2.2 the authors discuss the introduction of the Hv1 inhibitor 2GBI. Please, provide a brief explanation of how this compound interacts with the channel and clarify the significance of the ligand and how it mimics the effects of the ΔNΔC N264R mutant. .

Answer: We added some sentences on the bottom of page 9:

“In hHv1, the binding sites are determined by an interaction where the benzo ring points toward F150 in the transmembrane segment (S2). Additionally, the imidazole ring is positioned between the selectivity filter (D112, S1) and arginine R211 (R3, S4). Moreover, R211 interacts with the guanidinium group of 2GBI.”

 “In this way, 2GBI induces a voltage sensor trapping phenomenon like that produced by the ΔNΔC N264R variant. The decrease in the OFF-gating charge and the gating kinetics promoted by the inhibitor show the same pattern as the trapping induced by the ΔNΔC N264R mutant.”

  1. In Figure 5C, the linear correlation between the energy to move from an active to an inactive state and the hydrophobicity index is discussed. Please present a concise interpretation of the significance of this correlation in the context of voltage sensor activation.

Answer: We added some sentences on the bottom of page 12: “This may be a consequence of the fact that hydrophobicity can modify the transitions through which the voltage sensor is passing, affecting the activation energy of the channel as well as the displaced charge.”

  1. Please add a concise interpretation of the results presented in Figure 7, especially regarding the time course of ΔNΔC D160N N264R currents under ΔpH=2 conditions and explain the observed changes and their implications.

Answer: We added some sentences on the bottom of page 16: “Therefore, trapping can also be elicited by acidifying the inner vestibule of the channel near position 264 probably due to a transition in the kinetic model of voltage sensing highly dependent on intracellular pH”.

  1. In the discussion section, please consider adding more explanations of the results. For example, provide a brief explanation of why the monomeric CiHv1 is a relevant system for studying the properties of the Hv1 voltage sensor and explain why this system is suitable for exploring the relationships between the function and structure. Also, please clearly articulate the implications of the N264R mutation on gating currents. Discuss why the severe reduction in macroscopic currents and the observed behaviour of OFF-gating currents suggest charge trapping in the voltage sensor. The role of N264 position in Hv1 should be also disused in more detail. Give a brief description of the broader significance of the N264 position, not only in the movement of the voltage sensor but also in proton passage within Hv1. Discuss how alterations at this position affect current density, kinetics, and activation curve, and how these changes align with the state-dependent exposure to the solvent.

Answer: A new paragraph was added at the beginning of Discussion

The monomeric CiHv1 is a relevant system for studying the Hv1 voltage sensor's properties and how they relate to the functions of the channel and its structure. The monomeric (ΔNΔC) CiHv1 retains most of the characteristics reported for the wild-type dimer Hv1, such as high proton selectivity, sensitivity to ΔpH, voltage dependence, and inhibition by 2GBI. These processes are related to sections of the protein that are highly conserved. For example, the ΔNΔC D160N mutant almost completely isolates the proton current, allowing the voltage sensor movement to be detected. It is also known that changes in the ΔpH of the channel modify the active-resting equilibrium of the voltage sensor depending on the difference between the external and internal pH, thus also involving the arginines that make up the voltage sensing.

Additionally, ΔpH adjustments modify the kinetics and activation energy in Hv1 voltage sensor. Moreover, 2GBI provides insights into the spatial distribution of amino acids and the distances between critical elements such as the selectivity filter, certain arginines comprising the voltage sensor and phenylalanine related to the gating charge transfer centre; in addition, 2GBI effectively prevents the passage of protons. Also, the N264 position is crucial in understanding proton passage through Hv1 channels. Accessibility experiments with MTS reagents demonstrate that their binding to cysteine at this site significantly decreases current density. This suggests this position strongly influences the channel permeation pathway while preserving proton selectivity. Furthermore, these experiments demonstrate that the N264 residue in the S4 transmembrane segment (voltage sensor) can bind to the MTS reagents when exposed to the solvent in the open state. Substituting it with an arginine also leads to decreased proton conduction and activation kinetics, slow tail currents, and a leftward shift in activation energy compared to the wild type (Sakata, Carmona). Considering the information above, the ΔNΔC N264R mutation enables the recording of gating currents due to its fast activation kinetics and decreased proton current. However, two effects emerge: first, the behavior of the voltage sensor in the ON-gating current overlaps with the proton current, and second, there is a decrease in the OFF-gating charge upon returning to the proton reversal potential. The OFF-gating currents show only a component carrying a small fraction of the total charge observed in the ON-gating current component. This suggests that the gating particles must surmount a significant energy barrier to return to their resting state and point towards a charge-trapping state of the voltage sensor due to the N264R mutation. Therefore, N264R can be directly related to the permeation of protons and the movement of the voltage sensor when it is activated.

  1. In my opinion, the authors should point out the connections between the results of the study and their broader significance. In this way, it will be possible to clarify the implications of the results for understanding the function and structure of Hv1.

Please see in the new section of the manuscript: Conclusions.

We really appreciated all comments and suggestions made, particularly the ones that pertain to clarifying the implications of our results.

Reviewer 2 Report

Comments and Suggestions for Authors

              The manuscript „ Trapping Charges Mechanism in Hv1 Channels (CiHv1) ”, by Miguel Fernández, Juan J. Alvear-Arias, Emerson M. Carmona, Christian Carrillo, Antonio Pena-Pichicoi, Erick O. Hernandez-Ochoa, Alan Neely, Osvaldo Alvarez, Ramon Latorre, Jose A. Garate, and Carlos Gonzalez, highlights, at the molecular level, that N264R or a positive charge density within this area promotes an electrostatic repulsion on the gating charges, which facilitates an upwards movement of R258 and R261.  The manuscript should undergo major changes because it does not respect the scientific steps, in which the author's work is promoted, for example, the current paper does not have a section for Conclusions.  All my comments and suggestions are below.

1.      Page 4: Please make corrections for the following sentence, after equation 1: „...where z@@@ is the fractional displacement...”. Please try to use an accepted format for your variables.

2.      Page 4, Figure 1, graph E: Please explain the errors from the graph: why at low voltage, the errors are bigger, compared to higher voltage values?

3.      Page 5, the authors said: „...currents at the @pH=0 but with different...”. please check the manuscript for non-supported typing.

4.      Figure 2, graphs A, B, and C: please insert the axis, to enhance the cleanliness.

5.      Figure 2, graph E: Please explain the higher values of the errors, which do not have any trend. Do you have instabilities in your process?

6.      Page 6: inside the comment from Figure 2: the authors said „...pulse of the ON gating current at successive times of 2,7,12,17,17,22,27,32,37,42,47,52 seconds...”. Why does „17” appear twice? Is it a mistake, or not?

7.      Page 6: the authors said: „...mutation in a N@C@D160N background...”. Please check the supported format in order to express the wanted notation.

8.      Page 7, Figure 3, graph D: Please explain the reason for your errors with such high values.

9.      Please make changes to your manuscript, meaning that after the Introduction you should insert the Materials and Methods section, followed up by results, and finish with the conclusions.

10.   Please insert the Conclusion section. The authors should be aware that they cannot publish a scientific paper without a dedicated section for their experimental conclusions.

Author Response

Please find enclosed the revised version of our manuscript entitled “Trapping charges Mechanism in Hv1 channels (CiHv1)” by Miguel Fernández, Juan J Alvear-Arias, Emerson Carmona, Christian Carrillo, Antonio Peña-Pichicoi, Erick O. Hernández-Ochoa, Alan Neely, Osvaldo Alvarez, Ramon Latorre, Jose Garate, Carlos Gonzalez that we are resubmitting for review and publication as an article the special issue “State-of-the-Art Molecular Biology in Chile, 2nd Edition” of IJMS
The manuscript was changed according to the Reviewer 4’s comments and we have highlighted the changes in yellow on the text. We are grateful for the commitment to improve the quality of manuscript and hope. Hopefully this new version of the manuscript is suitable for the potential publication in IJMS.

Reponses to Reviewer #4:

  1. Page 4: Please make corrections for the following sentence, after equation 1: „...where z@@@ is the fractional displacement...”. Please try to use an accepted format for your variables. DONE

ANSWER: The typographical error was corrected

  1. Page 4, Figure 1, graph E: Please explain the errors from the graph: why at low voltage, the errors are bigger, compared to higher voltage values? .

Answer: It can be demonstrated that channel fluctuations show a maximum at the half-voltage of the gating charge-voltage curve. Fluctuations decrease at lower and higher voltages since voltage sensors are at rest at large negative voltages and all activated at large positive voltages. Errors, therefore, are minimal at both voltage extremes. However, a low probabilities of opening (i.e., low voltages), errors are larger since gating current

  1. Page 5, the authors said: „...currents at the @pH=0 but with different...”. please check the manuscript for non-supported typing.

ANSWER: The typographical error was corrected

  1. Figure 2, graphs A, B, and C: please insert the axis, to enhance the cleanliness. .

ANSWER: Current and time scales are given to the right of each figure

  1. Figure 2, graph E: Please explain the higher values of the errors, which do not have any trend. Do you have instabilities in your process? .

ANSWER: In the first few seconds of the voltage pulse protocol a very small amount of charge is displaced by the gating currents. If the baseline is not exactly at 0, possibly shifted slightly offset on the ordinate axis it could generate a shift and overestimate the value of Qon and make the ratio between the charges smaller than one.

  1. Page 6: inside the comment from Figure 2: the authors said „...pulse of the ON gating current at successive times of 2,7,12,17,17,22,27,32,37,42,47,52 seconds...”. Why does „17” appear twice? Is it a mistake, or not? .

ANSWER: Thanks for the comment, but it was just a typographic error.

  1. Page 6: the authors said: „...mutation in a N@C@D160N background...”. Please check the supported format in order to express the wanted notation. DONE

ANSWER: The typographical error was corrected

  1. Page 7, Figure 3, graph D: Please explain the reason for your errors with such high values. .

ANSWER: The origen of the large error  is, as in comment 2, that at lower voltages current are smaller and more noisy

9.Please make changes to your manuscript, meaning that after the Introduction you should insert the Materials and Methods section, followed up by results, and finish with the conclusions.

ANSWER: the manuscript has been changed according to journal instructions

  1. Please insert the Conclusion section. The authors should be aware that they cannot publish a scientific paper without a dedicated section for their experimental conclusions.

ANSWER: The manuscript was restructured according to journal instructions.

We appreciate your careful reading of our manuscript and pointing out all ways to improve it.

Round 2

Reviewer 1 Report

Comments and Suggestions for Authors

The authors have improved the manuscript significantly improving its quality. I have no further comments

Reviewer 2 Report

Comments and Suggestions for Authors

The authors have made the needed changes, and they took into account all the comments and suggestions provided by the reviewer. The scientific quality of the current manuscript has been improved, and it can be accepted for publication.